# Anti-Inflammatory and Cytotoxic Activities of Clerodane-Type Diterpenes

**DOI:** 10.3390/molecules28124744

**Published:** 2023-06-13

**Authors:** Rubria Marlen Martínez-Casares, Liliana Hernández-Vázquez, Angelica Mandujano, Leonor Sánchez-Pérez, Salud Pérez-Gutiérrez, Julia Pérez-Ramos

**Affiliations:** 1Departamento de Sistemas Biológicos, Universidad Autónoma Metropolitana-Xochimilco, Calzada del Hueso 1100, Coyoacán 04960, CDMX, Mexico; rmartinezc@correo.xoc.uam.mx (R.M.M.-C.);; 2Departamento de Atención a la Salud, Universidad Autónoma Metropolitana-Xochimilco, Calzada del Hueso 1100, Coyoacán 04960, CDMX, Mexico

**Keywords:** clerodane, neo-clerodane, anti-inflammatory, cytotoxic activities

## Abstract

The secondary metabolites of clerodane diterpenoids have been found in several plant species from various families and in other organisms. In this review, we included articles on clerodanes and neo-clerodanes with cytotoxic or anti-inflammatory activity from 2015 to February 2023. A search was conducted in the following databases: PubMed, Google Scholar and Science Direct, using the keywords clerodanes or neo-clerodanes with cytotoxicity or anti-inflammatory activity. In this work, we present studies on these diterpenes with anti-inflammatory effects from 18 species belonging to 7 families and those with cytotoxic activity from 25 species belonging to 9 families. These plants are mostly from the Lamiaceae, Salicaceae, Menispermaceae and Euphorbiaceae families. In summary, clerodane diterpenes have activity against different cell cancer lines. Specific antiproliferative mechanisms related to the wide range of clerodanes known today have been described, since many of these compounds have been identified, some of which we barely know their properties. It is very possible that there are even more compounds than those described today, in such a way that makes it an open field to discover. Furthermore, some diterpenes presented in this review have already-known therapeutic targets, and therefore, their potential adverse effects can be predicted in some way.

## 1. Introduction

Diterpenes are metabolites that come from isoprene units; these compounds can be classified according to their structure [1]. One type of diterpene is clerodanes, which are found in a wide range of plant species, especially those from the Labiatae, Euphorbiaceae and Verbenaceae families [2,3]; they have also been found in bacteria, fungi and marine sponges. This type of diterpene has been extensively studied due to many of them having biological activity [1,2,3,4]. For example, clerodin has anthelminthic activity [5]; salvinorin A is an agonist of κ-opioid receptor-serotonin-2A [6] with potential for use as a treatment in neuropsychiatric disorders [7]; tinosinenosides A–C show cytotoxicity effects against HeLa [8]; and columbin has anti-inflammatory and anticancer efficacy [4].

Clerodanes are secondary metabolites; when these compounds are obtained from plants, they are biosynthesized in the chloroplasts from geranylgeranyl pyrophosphate, producing a labdane-type precursor skeleton, which can be transformed to a halimane-type intermediate, and then converted to either *cis*- or *trans*- clerodanes [3] (Figure 1a).

Clerodanes are bicyclic diterpenoids with a fused ring of decalin structure (C_1_–C_10_) and a side chain of six carbons at C_9_. They are classified according to the configuration at the ring fusion and the substituents in C_8_ and C_9_ into four types: *trans*-*cis*, *trans*-*trans*, *cis*-*cis* and *cis*-*trans* (Figure 1b). About 25% have a *cis* ring fusion, and 75% have 5:10 *trans* ring fusion [9].

In this review, we have included clerodanes and *neo*-clerodanes and their enantiomers *ent*-*neo*-clerodanes (Figure 2). Additionally, carbons 12 to 16 are usually oxidized to diene, furan, lactone or hydrofurofuran, which give structural characteristics to clerodane [10].

Cancer is a global health problem and is currently one of the main causes contributing to premature death worldwide [11]. At the present time, even with the great advances in medicine in our understanding and treatment of cancer with multimodal therapies including immunotherapy, gene-targeted therapy, chemotherapy, hormonal therapy and cancer vaccines [12] against specific cell targets, there are needs that have not been covered. These include more effective therapies, with fewer adverse effects, but also therapies at a more affordable cost. Thus, there is still a need to investigate more effective and less toxic compounds. Most of the chemotherapeutic drugs (nearly 65%) that are used in current cancer treatment regimens were originally isolated from natural products or their derivatives such as plants or microorganisms [13]. For instance, paclitaxel, a diterpene isolated from *Taxus brevifolia* (yew trees), classified as a taxane, is used in the therapy of various types of cancers [14]. Other examples include anthracyclines derived from *Streptomyces* strains, among them being doxorubicin, bleomycin and many others [13]. The cytotoxic activity of several clerodanes in different cancer cell lines has been described [1].

On the other hand, inflammation is an immune response to different stimuli, such as pathogens such as viruses and bacteria, traumas and chemical irritants [15]; that is to say, inflammation is a protective response of the body against harmful stimuli. Additionally, long-term inflammation could lead to several symptoms, such as pain, fatigue, insomnia, depression and gastrointestinal problems [16]. Chronic inflammation is associated with diseases such as cancer, diabetes and arthritis [17]. The inflammatory response leads to the production of pro-inflammatory mediators, such as cytokines, serotonin, leukotrienes and histamine [18]. These mediators promote vascular permeability, leukocyte migration, blood vessel dilatation and pain. The anti-inflammatory activity of terpenes, such as carvacrol, some carotenes and diterpenes, such as clerodanes, and triterpenes, has been studied [2,19].

In this review, 158 clerodanes and 70 *neo*-clerodanes (**1**, **56**, **57**, **71**–**73**, **94**–**132**, **141**–**158**, **184**–**187**, **196**, **197** and **207**–**210**) with cytotoxic and anti-inflammatory activities reported from 2015 to February of 2023 were included. A total of 56 articles were found; in Table 1, the plants, family, collection place and part of the plants from which the clerodanes and *neo*-clerodanes were isolated are shown.

Clerodanes and *neo*-clerodanes with cytotoxic activity are shown in Table 2, and their structures are shown in Figure 3, Figure 4, Figure 5, Figure 6, Figure 7, Figure 8, Figure 9, Figure 10, Figure 11, Figure 12 and Figure 13.

Clerodanes and neo-clerodanes’ anti-inflammatory activities are summarized in Table 3, and their structures are shown in Figure 14, Figure 15, Figure 16, Figure 17, Figure 18 and Figure 19.

## 2. Discussion

This review discusses research from the last 8 years on clerodane and *neo*-clerodane diterpenes that exhibit cytotoxic and anti-inflammatory activities. It presents studies on these diterpenes with anti-inflammatory effects from 18 species belonging to 7 families and those with cytotoxic activity from 25 species belong to 9 families. These plants mostly belong to the Lamiaceae, Salicaceae, Menispermaceae and Euphorbiaceae families. They include 228 clerodanes and *neo*-clerodanes, of which, 140 have cytotoxic activity, 88 have anti-inflammatory activity and crassifolin Q-U (**49**–**53**), compounds **74**–**77** and (-)-hardwickiic acid (**91**) have both activities. Compound **75** and **77** were alone in including acute toxicity, but they did not indicate LD_50_.

### 2.1. Cytotoxic Activity

All clerodanes included in this review are oxygenated; 58% of them have at least one acetate group, 47% a hydroxyl group, 49% a ring of lactone and 22% a ring of furan as substituents. Additionally, it was found that three diterpenes isolated from *Sheareria nana* (**125**–**127**) have -OSO_3_H.

We found that 82 compounds out of 140 were evaluated using the MTT assay, which is broadly used to measure the cytotoxic effects of drugs on cancer cell lines, and it is considered a quantitative cytotoxicity analysis; the assay is used more often because in itself, it is relatively straightforward and provides benefits due to the ease of its utility.

Compared to standard cancer therapies, in vitro studies have shown the cytotoxic and antiproliferative properties of different clerodane compounds. The mechanisms involved include growth inhibition, apoptosis, interference with DNA synthesis and driving DNA fragmentation in many cancer cell lines of mesenchymal, epithelial and hematopoietic origin [1,3].

Some clerodane compounds inhibit growth in cancer cell lines. Anacolosins A–F (**3**–**8**) and corymbulosins X and Y (**9**–**10**) isolated from *Anacolosa clarkii* exhibit cytotoxic properties in four paediatric cancer types [21]. Caseakurzin B (**29**) and caseakurzin J (**34**) from *Casearia kurzii* were investigated in a lung epithelial carcinoma cell line; the former arrested the cell cycle at the G2/M phase and the second at the S phase. Obtained from the same plant, corymbulosin M (**25**), caseamembrin B (**26**) and caseamembrin U (**27**) were also cytotoxic in three types of cancer cell lines. Of note, corymbulosin M (**25**) was the most potent of them and apparently even more active than etoposide, and it was shown that it affects the cell cycle at the G0/G1 stage [28]. Kurzipene D (**38**), also obtained from *C. kurzii*, has a potent antiproliferative effect compared to other kurzipenes and affects proliferation at the S stage. Further, one in vivo study used a xenograft tumor model in zebra fish embryos; this compound suppressed tumor proliferation and migration comparable to etoposide [26]. Crassifolins Q-U (**49**–**53**) from *Croton crassifolius* inhibited angiogenesis in HUVECs, and crassifolin U (**53**) had the strongest activity in this model [32]. Notably, the antitumor properties of casearins have been shown using in vivo and ex vivo methods [30]. Epoxy clerodane diterpene (**139**) isolated from *Tinospora cordifolia* had cytotoxic activity, inhibiting MCF7 growth by regulating the expression of the functional genes Rb1 and Mdm2 [55].

Several specific antiproliferative mechanisms related to the wide range of clerodanes known today have been described, since many of these compounds have been identified, some which we barely know their properties. It is very possible that there are even more compounds than those described today, in such a way that makes it an open field to discover. However, it is important to mention that clinical studies are required to demonstrate their efficacy in the therapy of the current cancer pandemic, and demonstrating their safety is also of great importance.

### 2.2. Anti-Inflammatory Activity

A total of 45% of the clerodanes with anti-inflammatory activity have at least one hydroxyl, 69% compounds contain a ring of lactones, 50% a ring of furans and 26% an acetate group as substituents.

We found that 63 compounds reported to have anti-inflammatory activity were evaluated for nitric oxide inhibition with the Griess assay on RAW264.7 macrophages or BV-2-cell-stimulated-LPS, and the clerodanes **157**, **158**, **185**, **186** and **207** showed the best activity in this test with IC_50_ values of less than 2 µM. In this review, we found that in vivo studies have only been performed for hautriwaic acid (**196**) and nepetolide (**204**).

The anti-inflammatory activity of clerodane diterpenoids mediated by different mechanisms has been demonstrated in in vitro and in vivo animal models. Compounds **154**, **155**, **157** and **158** from *Callicarpa arborea* showed potent inhibitory effects against the NLRP3 inflammasome by inhibiting Casp-1 activation and IL-1β in reticulum cell sarcoma cells [59].

Clerodane **74**–**77** and **206** from extracts of *Polyalthia longifolia* seeds inhibit inflammation, blocking the synthesis of prostaglandins and leukotrienes through highly selective binding to cyclooxygenases (COX) 1 and 2 and 5-lipooxygenase (5-LOX), respectively, compared to the nonsteroidal anti-inflammatory drugs diclofenac and indomethacin [71]. In 2008, clerodane **206** was associated with the suppression of neutrophil respiratory burst and degranulation, and it is thought that it is mediated at least in part by the inhibition of calcium mobilization, AKT (protein kinase B) and p38 mitogen-activated protein kinase pathways [77]. Hautriwaic acid (**196**) from *Dodonaea viscosa* leaves, used for rheumatism, exhibited anti-inflammatory activity in a mouse ear edema model [66]. Clerodane compounds **164**–**175** from *Callicarpa hypoleucophylla* suppress superoxide anion generation and elastase release, inhibiting the function of human neutrophils [61]. *Trans*-crotonin inhibits dextran- and histamine-induced oedema [2].

Compounds derived from the *Scutellaria* genus have strong interactions with inducible nitric oxide synthase, and because of that, they inhibit nitric oxide production [72]. Five clerodane diterpenoids from *Croton crassifolius* roots, named crassifolins Q–U (**49**–**53**), reduced the levels of IL-6 and TNF-α in lipopolysaccharide-stimulated RAW 264.7 cells [32]. Compounds **211**–**213** from *Tinospora crispa* diminish the production of pro-inflammatory mediators (IL-1β, IL-6, TNF-α, iNOs, CCL12 and COX-2) [74].

## 3. Conclusions

In summary, clerodane diterpenes have activity against different cell cancer lines. Furthermore, some of the diterpenes presented in this review have already-known therapeutic targets, and therefore, their potential adverse effects can be predicted in some way, but the discovery of new compounds and new mechanisms remains to be seen. Anyway, the study of possible new therapies for inflammation continues to be important in order to expand the options for the treatment of inflammatory diseases that afflict the world.

More than 50% of clerodanes included in this review with cytotoxic activity contain acetate groups; on the other hand, 69% of the compounds with anti-inflammatory effects have a ring of lactone.

## Figures and Tables

**Figure 1 molecules-28-04744-f001:**
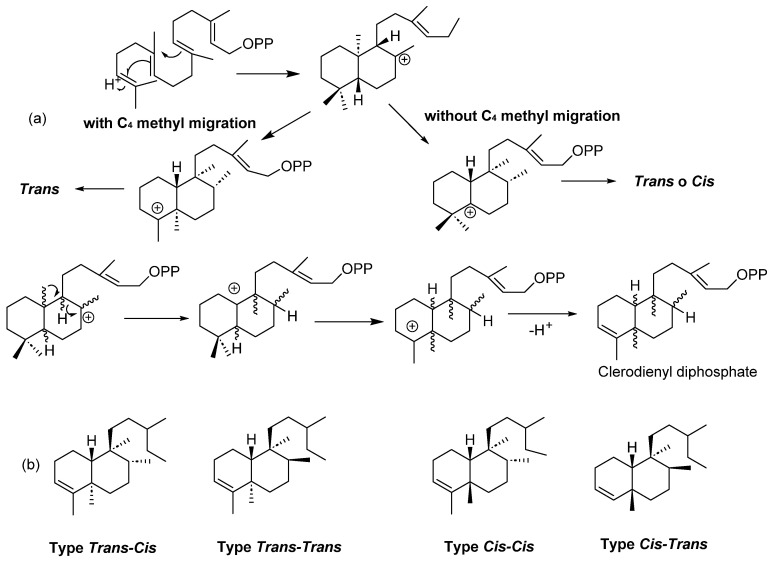
(**a**) Biosynthesis of clerodanes and (**b**) general structure of clerodanes.

**Figure 2 molecules-28-04744-f002:**
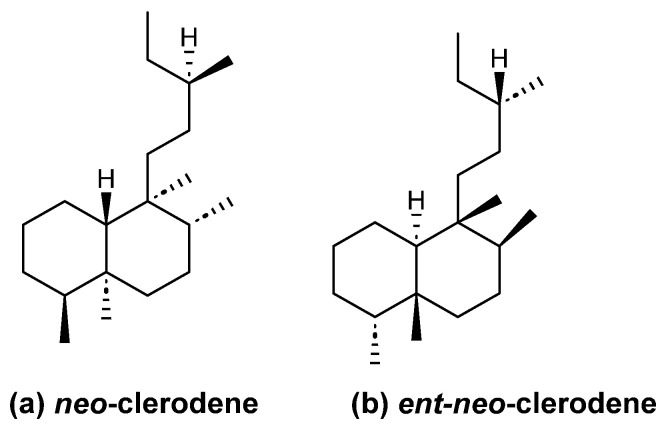
Absolute clerodane configuration.

**Figure 3 molecules-28-04744-f003:**
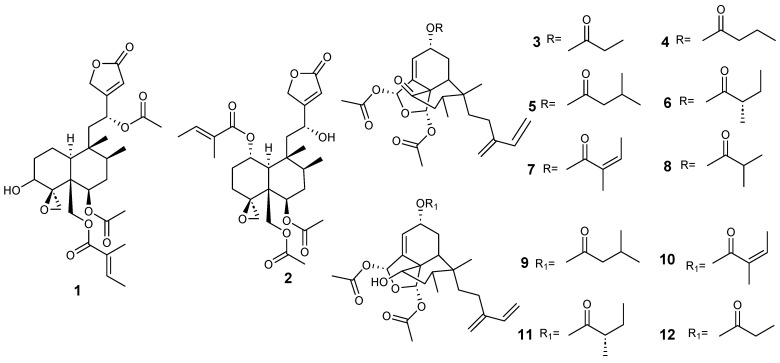
Isolated compound of *Ajuga decumbens* and *Anacolosa clarkii*.

**Figure 4 molecules-28-04744-f004:**
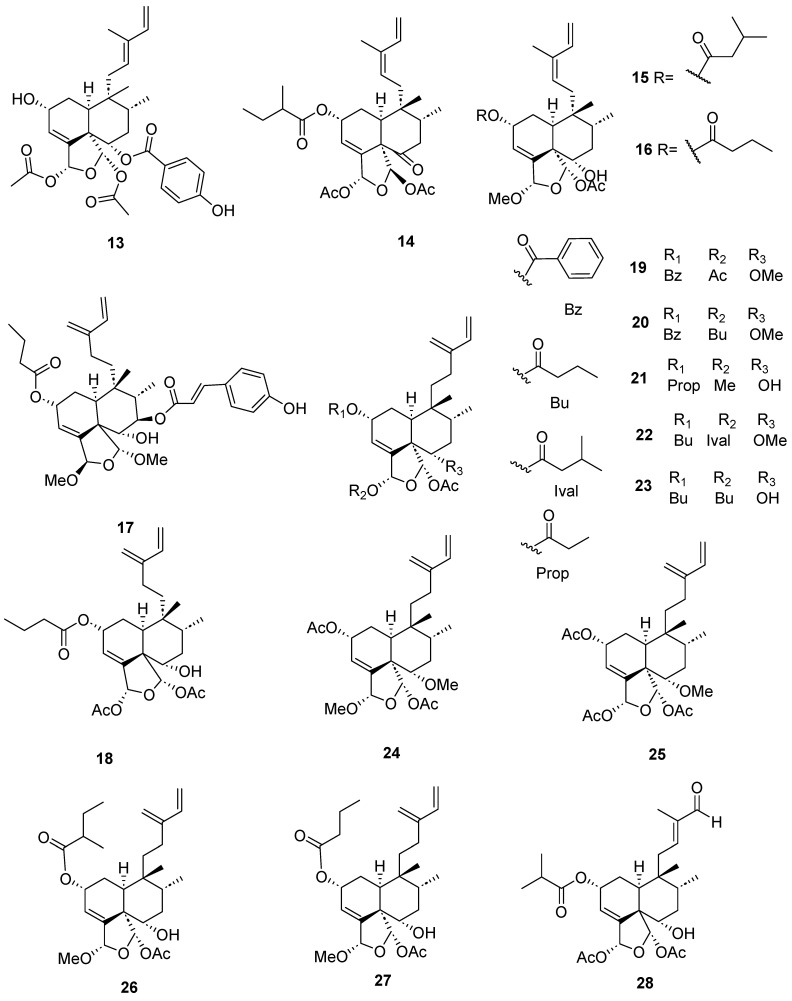
Isolated compounds of different species of *Casearia*.

**Figure 5 molecules-28-04744-f005:**
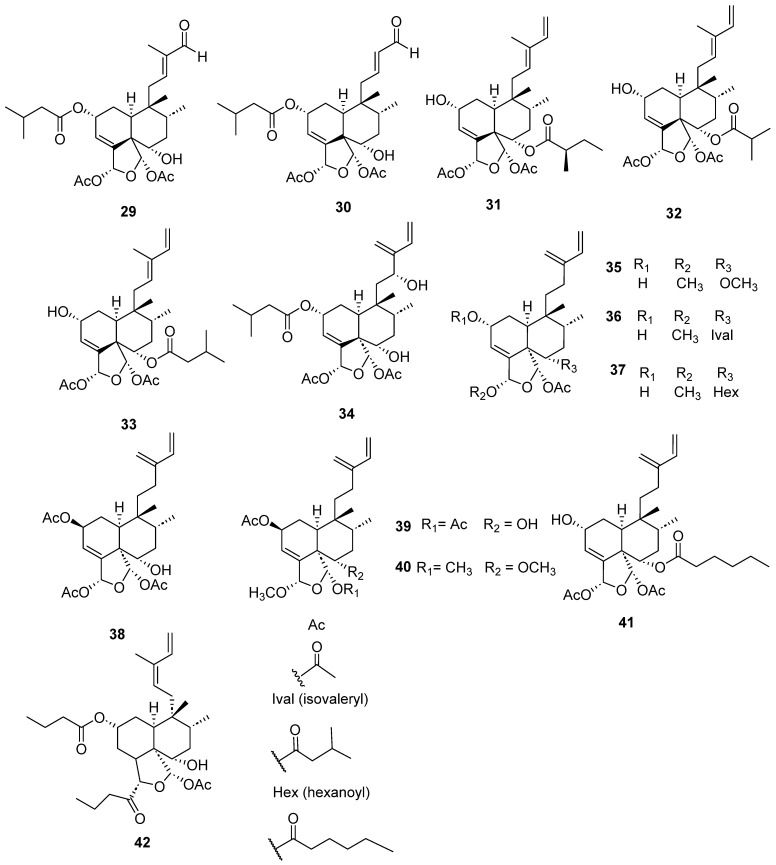
Isolated compounds of different species of *Casearia* (continued).

**Figure 6 molecules-28-04744-f006:**
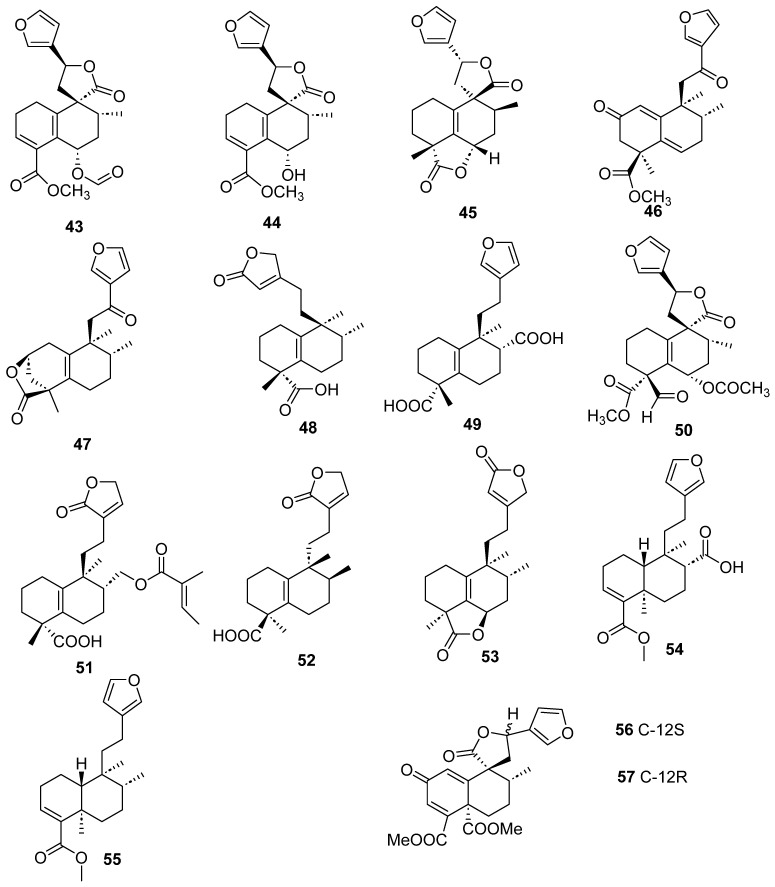
Isolated compounds of different species of *Croton*.

**Figure 7 molecules-28-04744-f007:**
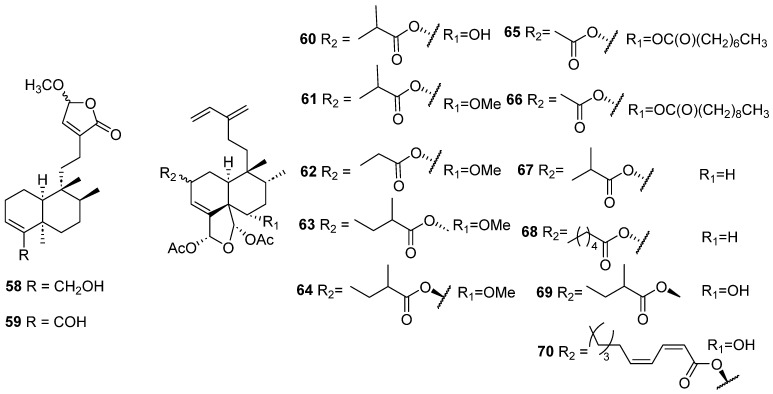
Isolated compounds of *Gottschelia schizopleura* and *Laetia corymbulosa*.

**Figure 8 molecules-28-04744-f008:**
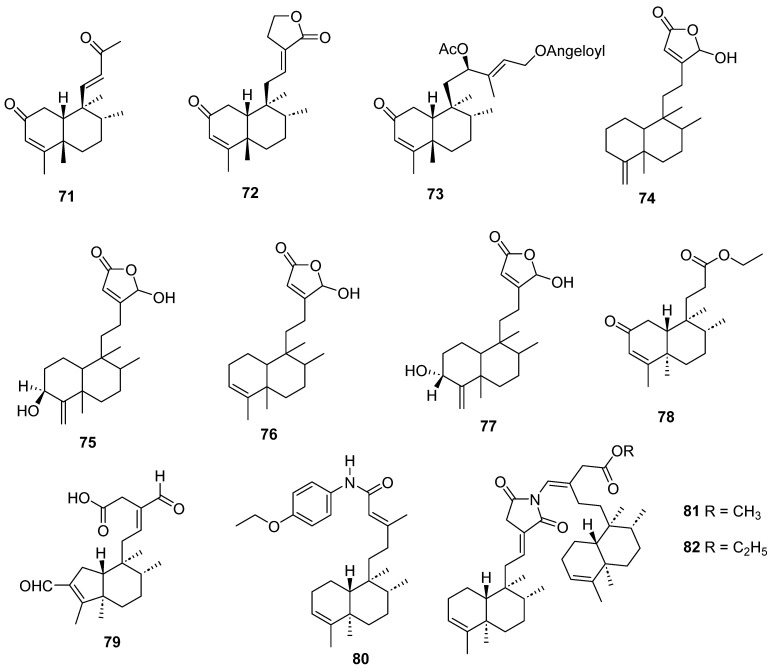
Isolated compounds of *Linaria japonica* and *Polyalthia longifolia*.

**Figure 9 molecules-28-04744-f009:**
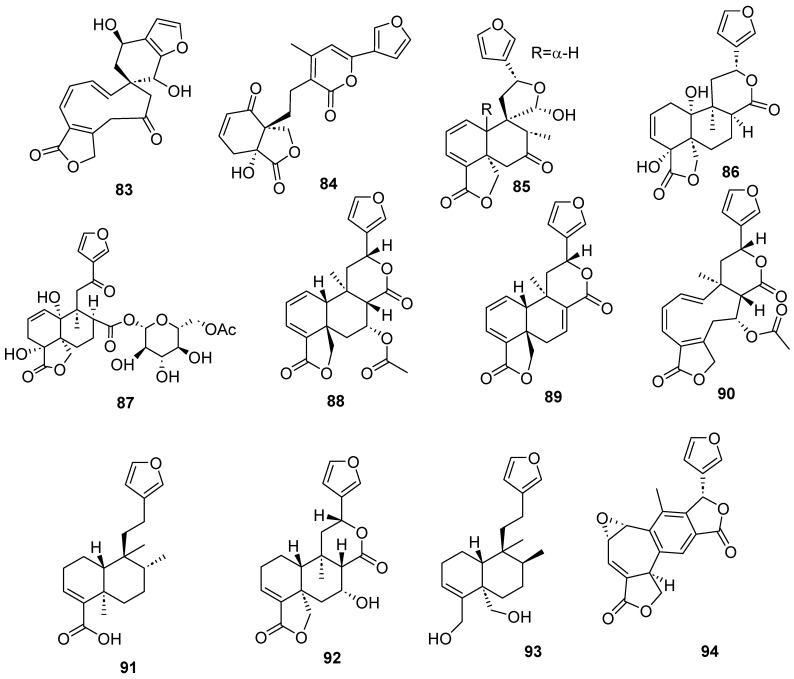
Isolated compounds of different species of *Salvia*.

**Figure 10 molecules-28-04744-f010:**
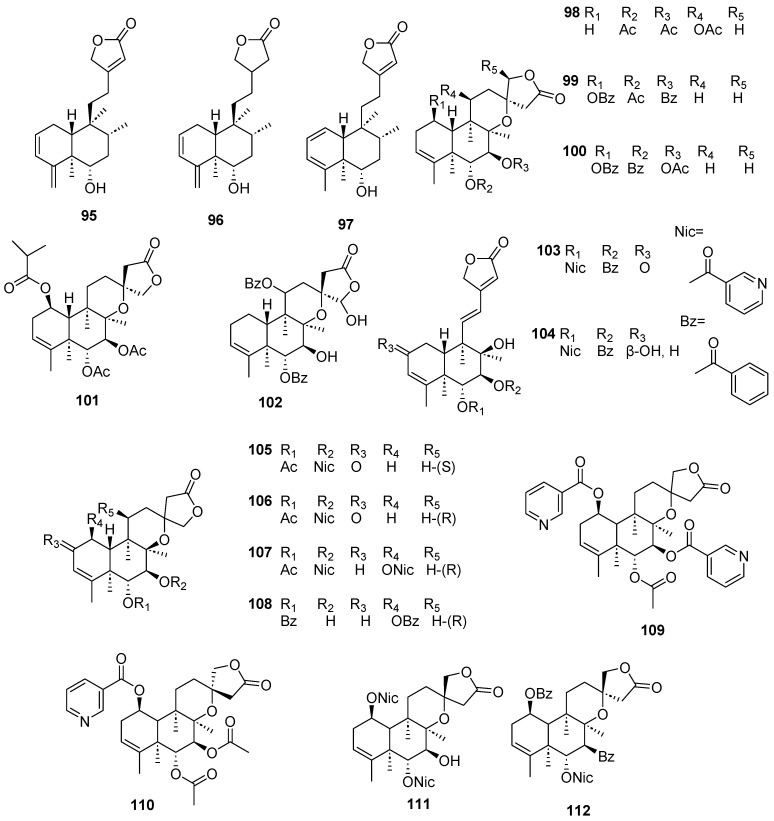
Isolated compounds of *Scutellaria barbata* (continued).

**Figure 11 molecules-28-04744-f011:**
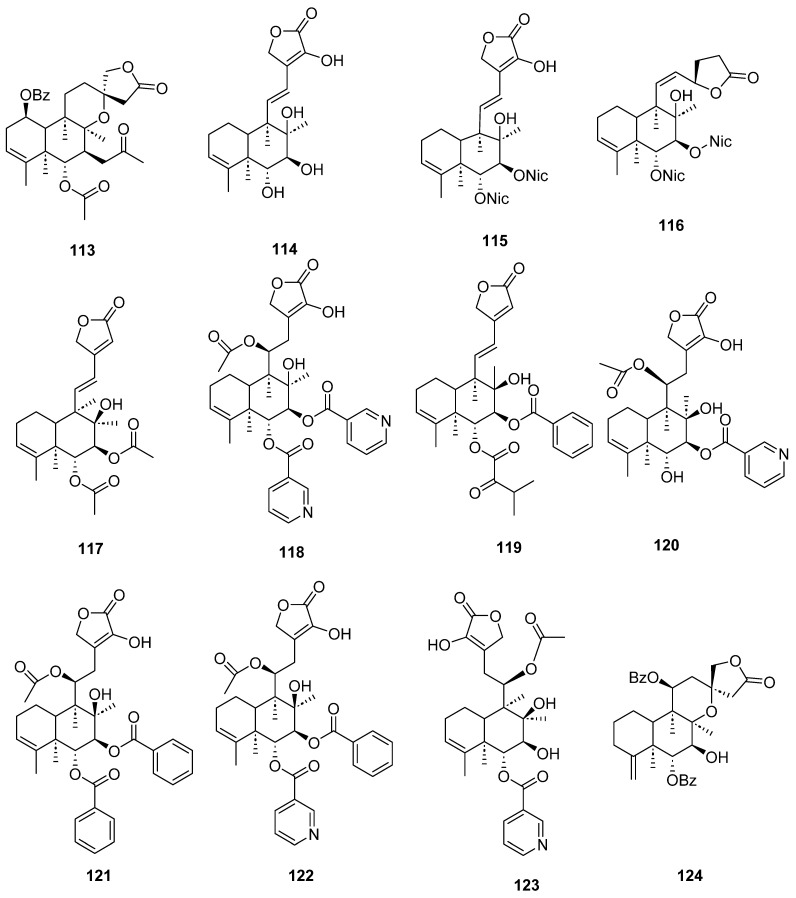
Isolated compounds of *Scutellaria barbata*.

**Figure 12 molecules-28-04744-f012:**
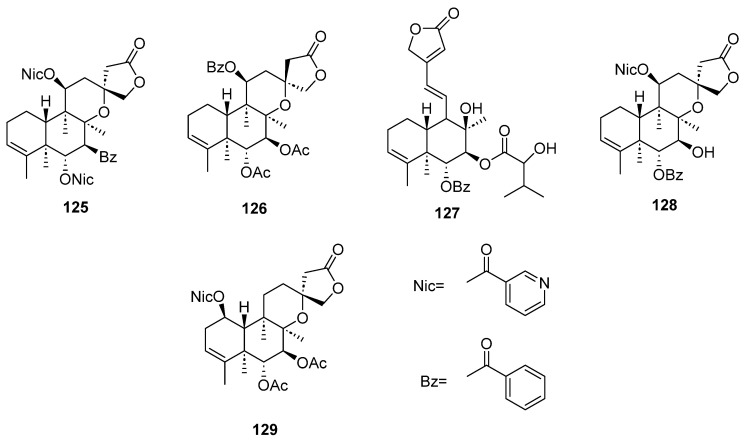
Isolated compounds of *Scutellaria strigillosa*.

**Figure 13 molecules-28-04744-f013:**
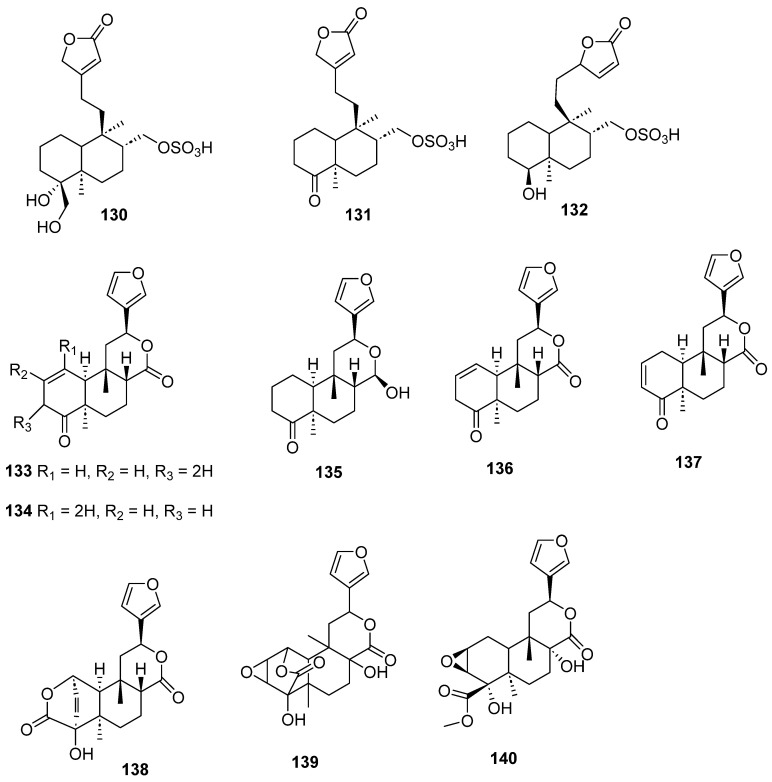
Isolated compounds of *Sheareria nana*, *Tinospora capillipes*, *Tinospora cordifolia* and *Tinospora sagittata*.

**Figure 14 molecules-28-04744-f014:**
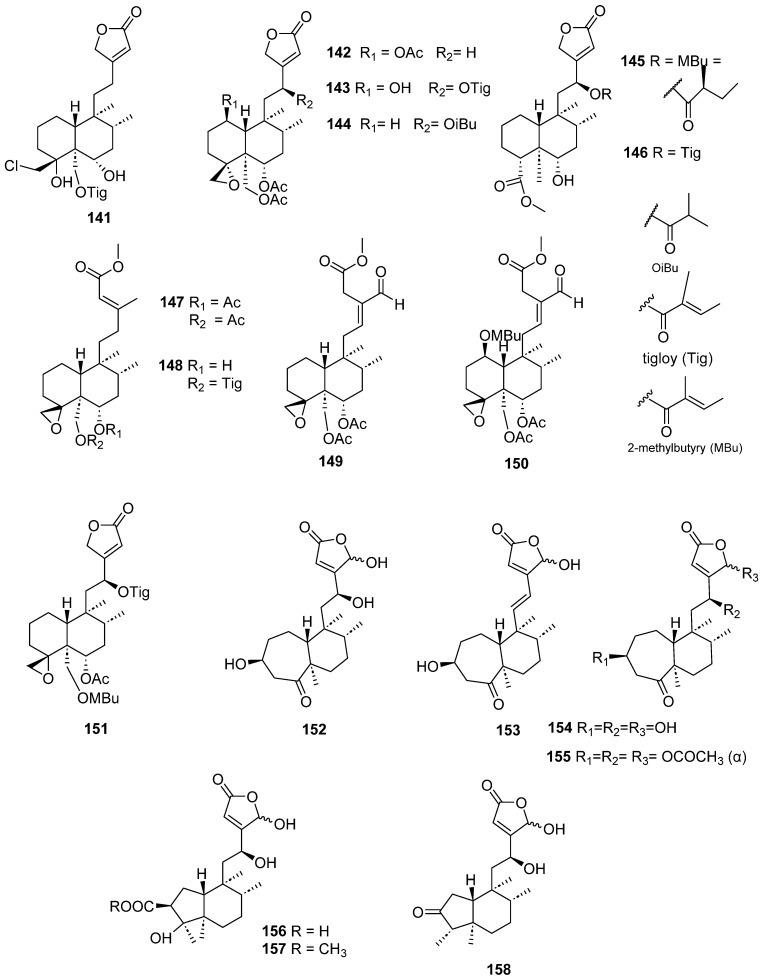
Compounds of *Ajuga pantantha* and *Callicarpa arborea* with anti-inflammatory activity.

**Figure 15 molecules-28-04744-f015:**
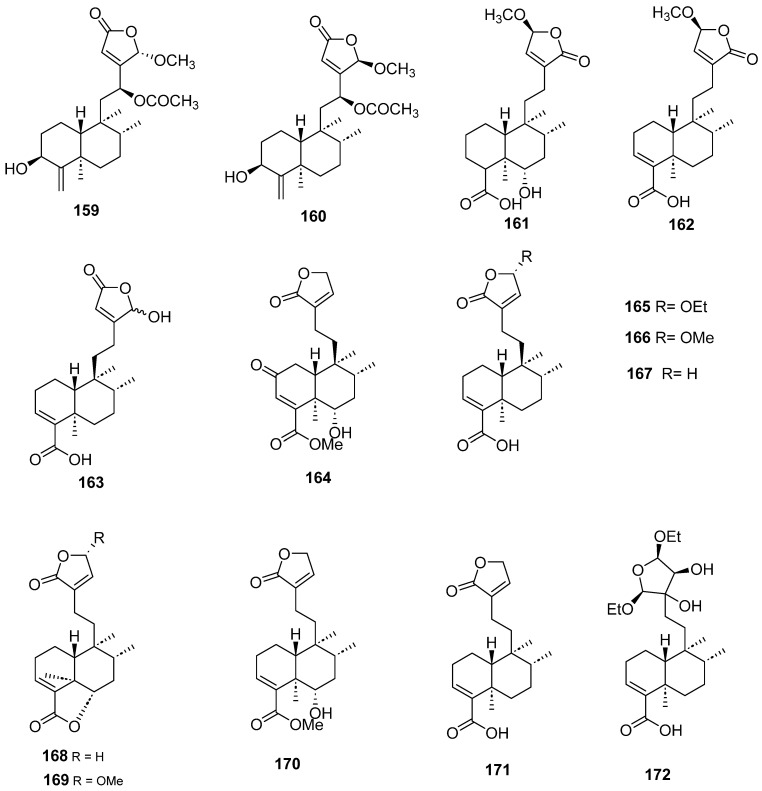
Compounds of *Callicarpa cathayana* and *Callicarpa hypoleucophylla* with anti-inflammatory activity.

**Figure 16 molecules-28-04744-f016:**
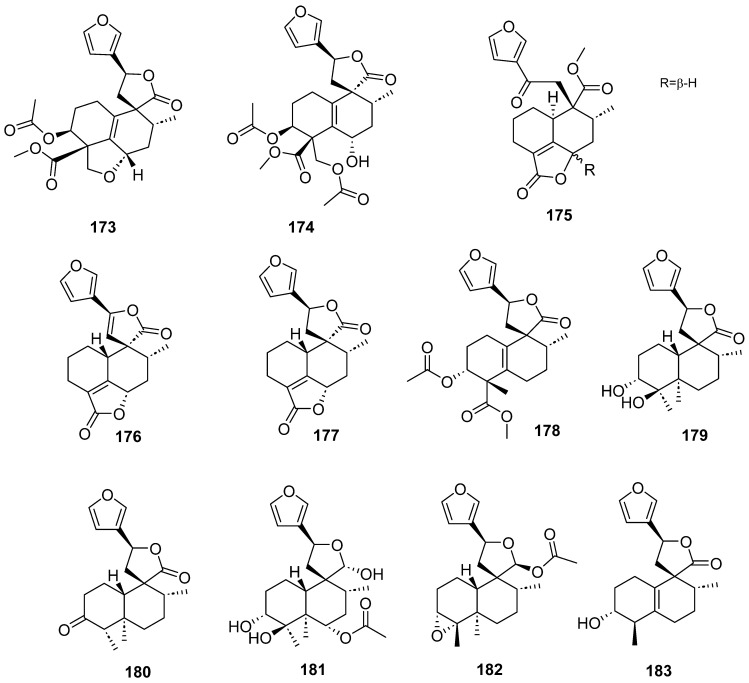
Compounds of different species of *Croton* with anti-inflammatory activity.

**Figure 17 molecules-28-04744-f017:**
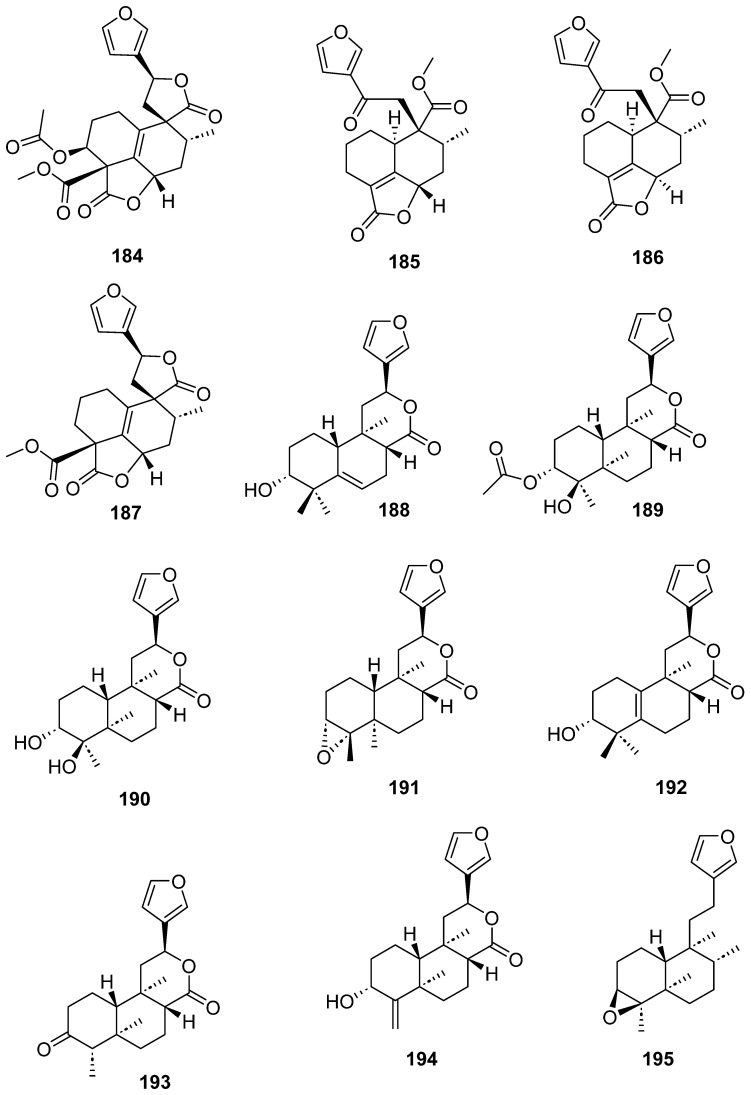
Compounds of different species of *Croton* with anti-inflammatory activity (continued).

**Figure 18 molecules-28-04744-f018:**
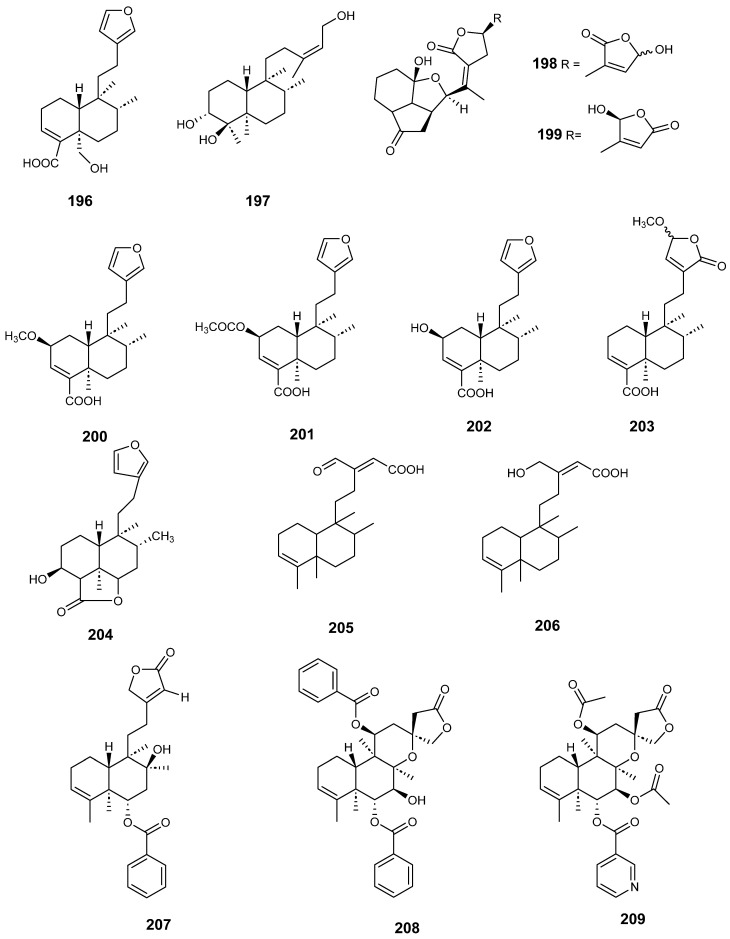
Compounds of *Dodonaea viscosa*, *Dysoxylum lukii*, *Jamesoniella autumnalis*, *Monon membranifolium*, *Nepeta suavis*, *Polyalthia longifolia* and *Scutellaria barbata* with anti-inflammatory activity.

**Figure 19 molecules-28-04744-f019:**
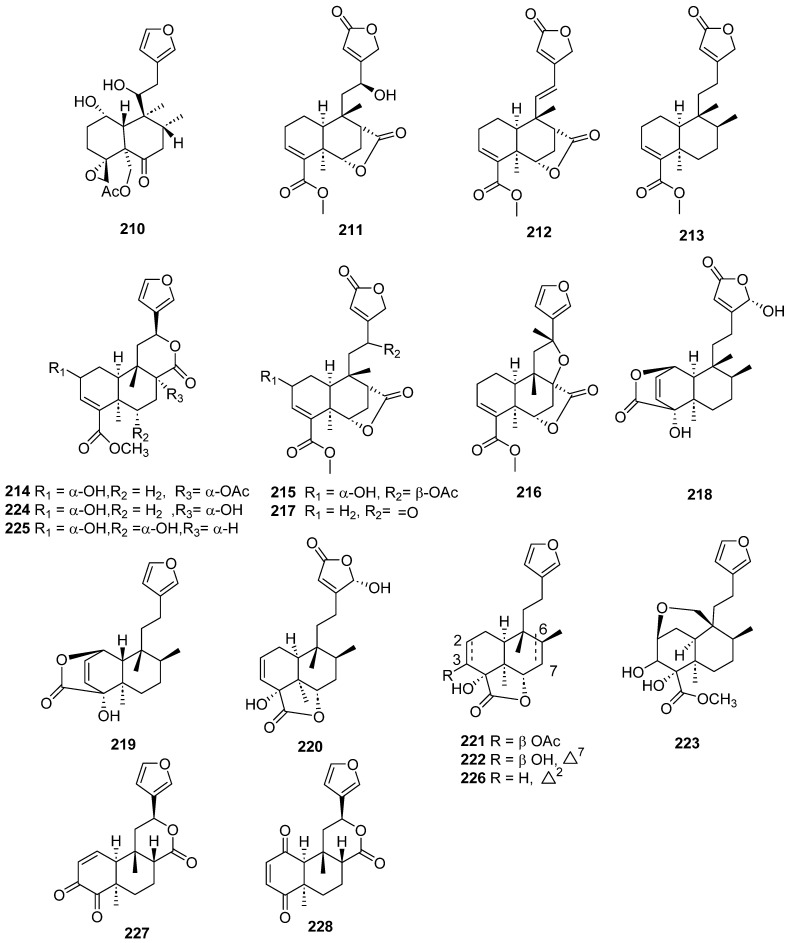
Compounds of different species of *Teucrium fructicans*, *Tinospora crispa* and *Tinospora sagittata* with anti-inflammatory activity.

**Table 1 molecules-28-04744-t001:** Part of plant, family and collection place of plants that contained clerodanes or *neo*-cleordanes.

Plant	Family	Part of Plant	Collection Place
*Ajuga decumbens* [20]	Lamiaceae	Aerial parts	Pingtan island of Fujian Province.
*Anacolosa clarkia* [21]	Olacaceae	Fruit, leaves and twigs of the plant	Bana Forest Preserve in Danang. NCI Natural Products Repository.
*Casearia corymbosa* [22]	Salicaceae	Stem bark	Othón P. Blanco, Quintana Roo, Mexico.
*Casearia graveolens* [23]	Salicaceae	Twigs	Chiang Rai Province, northern Thailand.
*Casearia grewiifolia* [24,25]	Salicaceae	Fresh fruits	Khon Kaen University campus.
		Leaves	Phu Loc–Thua Thien Hue, Vietnam.
*Casearia kurzii* [26,27,28,29]	Salicaceae	Fruit, leaves and twigs	Bana Forest Preserve in Danang, Vietnam.
		Twigs and leaves	Xishuangbanna County, Yunn an Province, P. R. China.
*Casearia sylvestris* [30]	Salicaceae	Leaves	Parque Estadual Carlos Botelho (São Miguel Arcanjo, São Paulo State.)
*Croton caudatus* [31]	Euphorbiaceae	Leaves and twigs	Xishuangbanna Prefecture, Yunnan Province, P. R. China.
*Croton crassifolius* [32,33,34]	Euphorbiaceae	Roots	Yulin City, Guangxi Province, China.
			Southeast China, Thailand, Vietnam, and Laos.
			Fujian Province, People’s Republic of China.
*Croton echioides* [35]	Euphorbiaceae	Stems	Brazil
*Croton oligandrus* [36]	Euphorbiaceae	Bark	Mount Eloundem, Central Region, Cameroon.
*Gottschelia schizopleura* [37]	Cephaloziellaceae	Aerial parts	Mount Alab, Sabah, North Borneo, Malaysia.
*Laetia corymbulosa* [38]	Salicaceae	Bark	The plant was provided by NCI/NIH (Frederick, MD, U.S.).
*Linaria japonica* [39]	Plantaginaceae	Whole plants	Hiroshima, Japan.
*Polyalthia longifolia* [40]	Annonaceae	Seeds	Tirupati, India.
*Polyalthia laui* [41]	Annonaceae	Roots	Hainan Province, China.
*Salvia amarissima* [42,43,44]	Lamiaceae	Leaves and flowers	Teotihuacan, State of Mexico.
		Aerial portions	Teotihuacan Valley
*Salvia involucrata* [45]	Lamiaceae	Aerial parts	Municipality of Xilitla, State of San Luis Potosí, Mexico.
*Salvia leucantha* [46]	Lamiaceae	Aerial parts	Yunnan Province, China.
*Scutellaria barbata* [47,48,49,50]	Lamiaceae	Whole plant	Linyi district, Shandong Province, China.
		Aerial parts	Purchased in a drugstore of Liaoning Guodayizhi Pharmaceutical Co., Ltd. China.
		Aerial parts	Purchased from Bozhou Herbal Market in Anhui Province, China
*Scutellaria strigillosa* [51,52]	Lamiaceae	Whole plants	Yantai district, Shandong Province, China.
		Whole plants	Hebei, Shandong, Zhejiang and Jilin Provinces, China
*Sheareria nana* [53]	Asteraceae	Whole herb	Jishou, Hunan Province, China.
*Tinospora capillipes* [54]	Menispermaceae	Whole herb	Xishuangbanna County, Yunnan Province, China.
*Tinospora cordifolia* [55]	Menispermaceae	Stems	India
*Tinospora sagittata* [56]	Menispermaceae	Roots	Anguo Medicine market in Hebei Province, China.
*Ajuga pantantha* [57,58]	Lamiaceae	Aerial parts	Yunnan Province, China.
		Aerial parts	Purchased from Anhui Province, China.
*Callicarpa arborea* [59]	Lamiaceae	Twigs	Xishuangbanna and Yuanyang Prefectures.
*Callicarpa cathayana* [60]	Lamiaceae	Dried aerial parts	Bozhou Herbal Market in Anhui Province, China.
*Callicarpa hypoleucophylla* [61]	Lamiaceae	Leaves and twigs	Kaohsiung city, Taiwan.
*Croton crassifolius* [32,62]	Euphorbiaceae	Roots	Guangxi Province, China.
*Croton floribundus* [63]	Euphorbiaceae	Roots	Provided by the company Mudas Nativas e Exóticas. LTDA of CNPJ, Araraquara Brazil.
*Croton laui* [64]	Euphorbiaceae	Leaves	Hainan Province, China.
*Croton poomae* [65]	Euphorbiaceae	Leaves and stems	Bung Kan Province, Thailand.
*Dodonaea viscosa* [66]	Sapindaceae	Leaves	Sierra de Huautla, Morelos State, Mexico.
*Dysoxylum lukii.* [67]	Meliaceae	Twigs and leaves	Xishuangbanna County, Yunnan Province, China.
*Jamesoniella autumnalis* [68]	Adelanthaceae	Whole plant	Wangtiane park, Changbaishan City, Jilin Province, China.
*Monoon membranifolium* [69]	Annonaceae	Twig extract	Thailand and Peninsula Malaysia.
*Nepeta suavis* [70]	Lamiaceae	Roots	Found in central and southern Europe, North Africa and southern Asia.
*Polyalthia longifolia* [71]	Annonaceae	Seeds	Seshachalam hills, Tirupati, India.
*Scutellaria barbata* [72]	Lamiaceae	Aerial parts	Baise city, Guangxi Province, China.
*Teucrium fructicans* [73]	Lamiaceae	Aerial parts	Jiansu Province, China.
*Tinospora crispa* [74,75]	Menispermaceae	Stems	Mengla County, Yunnan Province, China.
		Vines and leaves	Longzhou County, Guangxi Province, China.
*Tinospora sagittata* [76]	Menispermaceae	Tuberous roots	Shiyan city of Hubei Province, China.

**Table 2 molecules-28-04744-t002:** Clerodane diterpenes with cytotoxic activity.

Plant Source	Compound Name	Methods	Results	References
** *Ajuga* ** ** *decumbens* **	Compound **1**	CCK8 method A549 HeLa	IC_50_ µM 71.4 71.6	[20]
	Ajugamarin A1 (**2**)	A549 HeLa	76.7 5.39 × 10^−7^	
** *Anacolosa clarkii* **	Anacolosin A (**3**)	**SRB assay** A-673 SJCRH30 D283 Hep293TT	TGI μM 1.10 0.52 0.70 1.00	[21]
	Anacolosin B (**4**)	A-673 SJCRH30 D283 Hep293TT	1.00 0.50 0.60 0.90	
	Anacolosin C (**5**)	A-673 SJCRH30 D283 Hep293TT	1.10 0.67 0.66 1.00	
	Anacolosin D (**6**)	A-673 SJCRH30 D283 Hep293TT	1.20 0.73 0.66 0.80	
	Anacolosin E (**7**)	A-673 SJCRH30 D283 Hep293TT	3.10 1.90 2.00 1.80	
	Anacolosin F (**8**)	A-673 SJCRH30 D283 Hep293TT	4.10 2.30 2.30 3.20	
	Corymbulosin X (**9**)	A-673 SJCRH30 D283 Hep293TT	0.70 0.34 0.36 0.22	
	Corymbulosin Y (**10**)	A-673 SJCRH30 D283 Hep293TT	1.00 0.44 0.70 0.28	
	Compound **11**	A-673 SJCRH30 D283 Hep293TT	1.70 0.80 1.10 0.60	
	Caseamembrin S (**12**)	A-673 SJCRH30 D283 Hep293TT	0.90 0.36 0.50 0.30	
** *Casearia* ** ** *corymbosa* **	Casearborin c (**13**)	**SRB assay** HeLa SiHa Vero	CC_50_µM (SI) 13.44 77.36 50.26	[22]
** *Casearia graveolens* **	Caseariagraveolin (**14**)	**REMA assay** KB MCF-7	IC_50_ μM 2.48 6.63	[23]
** *Casearia grewiifolia* **	Caseargrewiin M (**15**)	**MTT assay** BT474 Chago-K1 Hep-G2 KATO-III SW620	IC_50_ µg/mL 6.30 6.10 4.64 5.50 5.50	[24,25]
	Caseargrewiin G (**16**)	BT474 Chago-K1 Hep-G2 KATO-III SW620	5.67 6.10 0.90 5.46 3.85	
	Caseagrewiifolin B (**17**)	**WST-1 assay** KB Hep-G2	**IC_50_** μM 6.2 7.0	
	Caseanigrescen D (**18**)	KB Hep-G2 LU-1 MCF-7 NIH-3T3	0.5 0.3 0.9 0.8 0.3	
** *Casearia* ** ** *kurzii* **	Kurziterpene A (**19**)	**MTT assay** A549, HeLa HepG_2_	IC_50_ μM 40.8 >60 >60	[26,27,28,29]
	Kurziterpene B (**20**)	A549 HeLa Hep-G2	19.7 12.1 49.3	
	Kurziterpene C (**21**)	A549, HeLa Hep-G2	>60 49.4 >60	
	Kurziterpene D (**22**)	A549, HeLa Hep-G2	18.3 9.0 >60	
	Kurziterpene E (**23**)	A549, HeLa Hep-G2	10.2 5.3 10.7	
		**Analysis via flow cytometry**	Apoptosis of HeLa	
	(2*R*,5*S*,6*S*,8*R*,9*R*,10*S*,18*S*,19*S*)-2,19-diacetoyloxy-6,18-dimethoxy-18,19-epoxycleroda-3,13(16),14-triene (**24**)	**MTT assay** A549 HeLa Hep-G2	IC_50_ μM >60 17.9 >60	
	Corymbulosin M (**25**)	A549 HeLa Hep-G2	5.5 4.1 9.3	
		**Analysis via flow cytometry**	Apoptosis of HeLa	
	Caseamembrin B (**26**)	**MTT assay** A549 HeLa Hep-G2	IC_50_ μM 36.1 18.8 >60	
	Caseamembrin U (**27**)	A549 HeLa Hep-G2	33.2 15.6 >60	
	Caseakurzin A (**28**)	**QIR assay** A549	IC_50_ μM 10.8	
	Caseakurzin B (**29**)	**QIR assay** A549	IC_50_ μM 4.4	
		**Cell apoptosis assay**	Apoptosis of A549	
	Caseakurzin C (**30**)	**QIR assay** A549	IC_50_ μM 30.3	
	Caseakurzin D (**31**)		27.8	
	Caseakurzin E (**32**)		32.7	
	Caseakurzin F (**33**)		26.8	
	Caseakurzin J (**34**)	**QIR assay** A549	IC_50_ μM 4.6	
		**Cell apoptosis assay**	Apoptosis of A549	
	Kurzipene A (**35**)	**MTT assay** Hep-G2 A549 HeLa K562	IC_50_ μM >60 >60 >60 >60	
	Kurzipene B (**36**)	Hep-G2 A549 HeLa K562	>60 32.6 54.6 >60	
	Kurzipene C (**37**)	Hep-G2 A549 HeLa K562	>60 >60 >60 >60	
	Kurzipene D (**38**)	Hep-G2 A549 HeLa K562	9.7 10.9 12.4 7.2	
		**Flow cytometry**	Apoptosis of Hep-G2	
		**Anti-tumor assay** using zebrafish model	It blocked tumor cell invasion and metastasis	
	Kurzipene E (**39**)	Hep-G2 A549 HeLa K562	>60 >60 >60 >60	
	Kurzipene F (**40**)	Hep-G2 A549 HeLa K562	>60 >60 33.1 >60	
	Corymbulosin V (**41**)	Hep-G2 A549 HeLa K562	16.8 11.2 14.2 10.3	
	Corymbulosin M (**25**)	Hep-G2 A549 HeLa K562	20.6 18.4 17.5 16.5	
** *Casearia* ** ** *sylvestris* **	Casearin X (**42**)	**Induced sarcoma tumor** 25 mg/kg/day	Tumor inhibition % 90.0	[30]
** *Croton* ** ** *caudatus* **	Crocleropene A (**43**)	**MTT assay** MCF-7	IC_50_ μM 35.8	[31]
	Crocleropene B (**44**)	MCF-7	40.2	
** *Croton* ** ** *crassifolius* **	Crassifolius A (**45**)	**Morphology**	Induced apoptosis	[32,33,34]
		**Western blot**	Caspase activation	
		**MTT assay** Hep3B Hep-G2	IC_50_ µM 17.91 42.04	
	Crassifolin C (**46**)	Hep-G2	51.63	
	Compound **47**	Hep-G2	45.22	
	Crassifolin B (**48**)	CT26.WT	96.6	
	Crassifolin Q (**49**)	**HUVEC assay**	Compounds **49**–**51** and **53** inhibited angiogenesis	
	Crassifolin R (**50**)			
	Crassifolin S (**51**)			
	Crassifolin T (**52**)	**HUVEC assay**	Anti-angiogenesis effect	
	Crassifolin U (**53**)	**HUVEC assay** Junction densities Vessel areas Vessel lengths	IC_50_ μM 7.20 48.27 8.62	
** *Croton* ** ** *echioides* **	CEH-1 (**54**)	**MTT assay** HTC	Compound **54** diminished 67% cell viability and 55 < 76%.	[35]
	CEH-4 (**55**)			
** *Croton* ** ** *oligandrus* **	Megalocarpoidolide D (**56**)	**MTT assay** A549 MCF-7	IC_50_ µM 63.8 136.2.	[36]
	12-epi-megalocarpodolide D (**57**)	A549 MCF-7	138.6 171.3	
** *Gottschelia schizopleura* **	Schizopleurolide A (**58**)	**MTT assay** HL-60 B16-F10	IC_50_ µM 38.47 47.25	[37]
	Schizopleurolide B (**59**)	HL-60 B16-F10	36.13 44.33	
** *Laetia corymbulosa* **	Corymbulosin I (**60**)	**Flow cytometry**	Compounds **60**, **61**, **12** and **11** induced apoptosis in MDA-MB-231	[38]
		**SRB assay** A549 MDA-MB-231 MCF-7 KB KB-VIN	IC_50_ µM 0.66 0.48 0.68 0.56 0.98	
	Corymbulosin K (**61**)	A549 MDA-MB-231 MCF-7 KB KB-VIN	0.47 0.49 0.50 0.45 0.49	
	Corymbulosin L (**62**)	A549 MDA-MB-231 MCF-7 KB KB-VIN	4.60 4.95 4.94 5.19 4.92	
	Corymbulosin N (**63**)	A549 MDA-MB-231 MCF-7 KB KB-VIN	5.04 4.90 5.82 5.23 5.19	
	Corymbulosin O (**64**)	A549 MDA-MB-231 MCF-7 KB KB-VIN	4.75 3.31 4.65 4.25 4.76	
	Corymbulosin P (**65**)	A549 MDA-MB-231 MCF-7 KB KB-VIN	5.98 4.93 6.39 5.16 5.03	
	Corymbulosin Q (**66**)	A549 MDA-MB-231 MCF-7 KB KB-VIN	40.2 20.5 31.7 19.8 39.2	
	Corymbulosin S (**67**)	A549 MDA-MB-231 MCF-7 KB KB-VIN	>40 22.9 26.2 25.1 26.6	
	Corymbulosin T (**68**)	A549 MDA-MB-231 MCF-7 KB KB-VIN	2.29 0.49 0.69 0.56 0.61	
	Corymbulosin V (**41**)	A549 MDA-MB-231 MCF-7 KB KB-VIN	4.76 4.73 5.19 4.74 4.88	
	Caseamembrin S (**12**)	A549 MDA-MB-231 MCF-7 KB KB-VIN	0.58 0.45 0.66 0.53 0.90	
	Caseamembrin E (**69**)	A549 MDA-MB-231 MCF-7 KB KB-VIN	0.53 0.40 0.55 0.43 0.51	
	Corymbulosin A (**70**)	A549 MDA-MB-231 MCF-7 KB KB-VIN	0.45 0.43 0.44 0.42 0.45	
	Compound **11**	A549 MDA-MB-231 MCF-7 KB KB-VIN	4.15 0.54 0.89 0.73 4.07	
** *Linaria* ** ** *japonica* **	Linarenone C (**71**)	**MTT assay** A549	IC_50_ µM 51.2	[39]
	Linarenone E (**72**)		86.5	
	Linarienone (**73**)		79.0	
** *Polyalthia longifolia* **	16-hydroxy-cleroda-4(18),13-dien-16,15-olide (**74**)	**Evaluation of morphometric liver and biochemical parameters in (NDEA+PB)-induced HCC rats**	Compound **75** and **77** restored the parameters’ biochemical and liver morphology	[40]
		**MTT assay** Hep-G2	**IC_50_** µg/mL 34.33	
	3α,16α-dihydroxy-cleroda-4(18),13(14)Z-dien-15,16-olide (**75**)	Hep-G2 HuH-7	14.34 47.32	
	16α-hydroxy-cleroda-3,13(14)Z-dien-15,16-olide (**76**)	Hep-G2	29.21	
	3β-16a-dihydroxy-cleroda-4(18),13(14)Z-dien-15,16-olide (**77**)	Hep-G2 HuH-7	24.91 48.57	
* **Polyalthia laui** *	Polylauiester A (**78**)	**MTT assay** HeLa MCF-7 A549	IC_50_ μM 34.84 33.21 35.65	[41]
	(4→2)-*abeo*-2,13-diformyl-cleroda-2,12*E*-dien-14-oic acid (**79**)	HeLa MCF-7 A549	39.31 37.35 37.82	
	Polylauiamide B (**80**)	HeLa MCF-7 A549	28.09 29.16 29.25	
	Polylauiamide C (**81**)	HeLa MCF-7 A549	25.01 30.30 28.65	
	Polylauiamide D (**82**)	HeLa MCF-7 A549	26.73 27.03 28.88	
** *Salvia* ** ** *amarissima* **	Teotihuacanin (**83**)	**SRB assay** MDA-MB-231 HeLa HCT-15 HCT-116 MCF-7	IC_50_ μM 12.3 13.7 12.9 10.9 >20	[42,43,44]
	Amarissinin A (**84**)	MCF-7 MCF-7/Vin^+^ MDA-MB-231 HeLa	18.2 0.27 19.3 14.0	
	Amarissinin B (**85**)	**SRB assay**	**83**, **84**, **85**, **86** and **87** exhibited MDR modulatory effects in mammalian cancer cells	
	Amarissinin C (**86**)			
	Amarisolide F (**87**)	**SRB assay** MCF-7 HeLa HCT-15 HCT-116 MDA-MB-231	IC_50_ μM 42.1 >42 >42 >42 >42	
** *Salvia involucrata* **	Involucratin A (**88**)	U251 PC-3 K562 SKLU-1	49.6 14.7 24.8 12.6	[45]
	Involucratin B (**89**)	U251 PC-3 K562 HCT-15 MCF-7 SKLU-1 COS-7	5.1 23.5 34.7 11.8 0.5 36.7 21.6	
	Involucratin C (**90**)	PC-3 K562 HCT-15 SKLU-1 COS-7	11.0 19.4 9.7 16.8 11.9	
	(-)-Hardwickiic acid (**91**)	U251 PC-3 K562 HCT-15 MCF-7 SKLU-1 COS-7	22.4 1.8 45.5 10.4 1.4 11.5 19.8	
	7α-hydroxybacchotricuneatin A (**92**)	U251 PC-3 K562 HCT-15 SKLU-1 COS-7	3.8 12.8 20.2 13.3 33.0 14.2	
	Kingidiol (**93**)	**SRB assay** U251 PC-3 K562 HCT-15 MCF-7 SKLU-1 COS-7	IC_50_ μM 22.4 13.0 51.6 15.5 0.8 22.9 19.7	
** *Salvia* ** ** *leucantha* **	Salvileucantholide (**94**)	**MTT assay** HCT-116 BT474 HepG2 Hsp90	IC_50_ µM 32.61 25.02 37.35 6.78	[46]
** *Scutellaria barbata* **	Scubatine A (**95**)	**MTT assay** HL-60 A549	IC_50_ µM >20 >20	[47,48,49,50]
	Scubatine B (**96**)	HL-60 A549	>20 >20	
	Scubatine C (**97**)	HL-60 A549	>20 >20	
	Scubatine D (**98**)	HL-60 A549	>20 >20	
	Scubatine E (**99**)	HL-60 A549	>20 >20	
	Scubatine F (**100**)	HL-60 A549	15.3 10.4	
	Scutebata E (**101**)	**MTT assay** HL-60 A549 LoVo	IC_50_ µM >20 >20 61.23	
	Scutolide K (**102**)	HL-60 A549	>20 >20	
	Scutebata X (**103**)	SGC-7901 MCF-7 A549	>40 37.2 >40	
	Scutebata Y (**104**)	SGC-7901 MCF-7 A549	>40 >40 >40	
	Scutebata Z (**105**)	SGC-7901 MCF-7 A549	>40 >40 >40	
	Scutebata A_1_ (**106**)	SGC-7901 MCF-7 A549	>40 >40 35.5	
	Scutebata B_1_ (**107**)	SGC-7901 MCF-7 A549	>40 >40 >40	
	Scutebata C_1_ (**108**)	SGC-7901 MCF-7 A549	17.9 29.9 35.7	
	Barbatin H. (**109**)	LoVo MCF-7 SMMC-7721 HCT-116	32.44 49.86 48.75 44.24	
	Scuterbarbatine F (**110**)	LoVo MCF-7 SMMC-7721 HCT-116	23.32 49.19 58.12 78.83	
	6-*O*-nicotinoylscutebarbatine G (**111**)	LoVo SMMC-7721 HCT-116	29.44 65.51 54.44	
	Scutebata G (**112**)	LoVo MCF-7 SMMC-7721 HCT-116	22.56 31.33 32.49 28.29	
	Scutebata D (**113**)	LoVo MCF-7 SMMC-7721 HCT-116	20.75 31.42 29.24 62.66	
	Barbatin C (**114**)	LoVo MCF-7 SMMC-7721 HCT-116	37.99 28.06 72.69 32.94	
	Scutebarbatine A (**115**)	LoVo	67.77	
	Scutebarbatine G (**116**)	LoVo SMMC-7721 HCT-116	56.46 70.16 44.25	
	6,7-di-*O*-acetoxybarbatin A (**117**)	LoVo MCF-7 SMMC-7721 HCT-116	60.33 37.31 77.93 32.28	
	Scutebarbatine X (**118**)	LoVo MCF-7 SMMC-7721 HCT-116	43.21 74.83 46.14 62.11	
	Barbatin F (**119**)	LoVo HCT-116	56.46 44.25	
	Barbatin G (**120**)	LoVo SMMC-7721 MCF-7 HCT-116	60.33 37.31 77.93 32.28	
	Scutebata A (**121**)	LoVo SMMC-7721 MCF-7 HCT-116 HL-60 A549	4.57 7.68 5.31 6.23 >20 >20	
	Scutebata B (**122**)	LoVo SMMC-7721 MCF-7 HCT-116	10.73 18.96 10.27 28.48	
	Scutebata C (**123**)	LoVo SMMC-7721 MCF-7	47.15 33.18 38.79	
	Scutebata P (**124**)	LoVo SMMC-7721 MCF-7 HCT-116 HL-60 A549 HCT-116	15.17 42.63 32.49 23.97 5.6 21.7 23.97	
** *Scutellaria strigillosa* **	Scutestrigillosin A (**125**)	**REMA assay** P-388 HONE-1, HT-29 MCF-7	IC_50_ μM 5.8 3.5 4.7 5.7	[51,52]
	Scutestrigillosin B (**126**)	P-388 HONE-1 HT-29 MCF-7	5.2 4.2 4.1 6.0	
	Scutestrigillosin C (**127**)	P-388 HONE-1, HT-29 MCF-7	7.1 3.9 6.4 7.7	
	Scutestrigillosin D (**128**)	P388 HONE 1 HT-29 MCF-7	5.6 3.4 4.7 5.2	
	Scutestrigillosin E (**129**)	P388 HONE 1 HT-29 MCF-7	8.9 7.3 8.1 7.4	
** *Sheareria nana* **	Sheareria A (**130**)	**CCK8 assay** HeLa PANC-1 A549	IC_50_ µM 11.6 7.1 9.3	[53]
	Sheareria B (**131**)	HeLa PANC-1 A549	9.4 5.6 6.8	
	Sheareria C (**132**)	HeLa PANC-1 A549	17.2 9.8 12.5	
** *Tinospora cordifolia* **	Tinocapillin A (**133**)	**MTT assay** A549 HepG2 HeLa OS-RC-2	IC_50_ µM 14.0 9.9 9.7 10.6	[54]
	Tinocapillin B (**134**)	A549 HepG2 HeLa OS-RC-2	9.6 10.1 12.0 19.1	
	Tinocapillin C (**135**)	A549 HeLa	53.2 67.5	
	Tinocallone A (**136**)	A549 HepG2 HeLa	67.8 68.4 79.3	
	Tinocallone C (**137**)	A549 HepG2 HeLa OS-RC-2	16.3 13.8 17.5 12.8	
	Columbin (**138**)	A549 HeLa	77.3 58.4	
** *Tinospora capillipes* **	ECD (epoxy clerodane diterpene) (**139**)	**MTT assay** V79 MCF-7 Vero	IC_50_ µM 52.7 3.2 45.8	[55]
		**qPCR analysis**	Inhibited MCF-7 grow by regulation the expression of genes such Cdkn2A, Rb1, Mdm2 y p53	
** *Tinospora sagittata* **	Tinosporin A (**140**)	**MTT assay** HL-60 MCF-7	IC_50_ µM 18.63 23.58	[56]

Compound **1** (1*S*,4*aS*,5*R*,6*S*,8*R*,8*aS*)-8-acetoxy-5-((*R*)-2-acetoxy-2-(5-oxo-2,5-dihydrofuran-3-yl)ethyl)-2-hydroxy-5,6-dimethyloctahydro-8aH-spiro[naphthalene-1,2′-oxiran]-8a-yl)methyl (E)-2-methylbut-2-enoate; Compound **11** (2*R*,5*S*,6*S*,8*R*,9*R*,10*S*,18*R*,19*S*)-18,19-di-*O*-acetyl-18,19-epoxy-6-hydroxy-2-(2′-methylbutanoyloxy)cleroda-3,13-(16),14triene; Compound **47** 6-[2-(furan-3-yl)-2-oxoethyl]-1,5,6-trimethyl-10-oxatricyclo[7.2.1.02,7] dodec-2(7)-en-11-one. 3-[4,5-dimethylthiazol-2-yl]-2,5 diphenyl tetrazolium bromide (MTT); sulforhodamine B (SRB); *N*-nitrosodiethylamine and phenobarbital sodium (NDEA+PB); cell counting kit 8 assay (CCK8); resazurin microplate assay (REMA); protein 90 kDa of family of chaperones (Hsp90); concentration cytotoxic at 50% (CC_50_); quinone reductase assay (QIR); selective index (SI); total growth inhibitory (TGI); breast cancer (MCF-7); breast cancer resistant at vinblastine (MCF-7/Vin); breast ductal carcinoma (BT474); cervix adenocarcinoma (HeLa); cervix squamous carcinoma (SiHa); colon adenocarcinoma (SW620, HCT-15, HCT-116 and HT-29) colon cancer (LoVo); chronic myeloid leukemia (K562); epidermoid carcinoma of the nasopharynx (KB); Ewing sarcoma (A-673); gastric carcinoma (KATO-III, SGC-7901); glioblastoma (U251); hepatocarcinoma (Hep293TT, Hep3B, Hep-G2, SMMC-7721, HCC, HuH-7); human umbilical vein endothelial cells (HUVEC); liver tumor cells of *Rattus norvegicus* (HTC); lymphoma cells (P388); lung adenocarcinoma (LU-1, SKLU-1, A549); medulloblastoma (D283); mouse colon adenocarcinoma (CT26.WT); mouse embryonic fibroblast cell line (NIH-3T3); musculus skin melanoma (B16-F10); normal green monkey kidney cell line (Vero); normal monkey kidney (COS-7); normal prostate epithelium (PNT2); promyelocytic leukemia (HL-60); prostate cancer (PC-3); P-gp-overexpressing MDR subline of KB (KB-VIN); pancreatic carcinoma (PANC-1); renal carcinoma (OS-RC-2); rhabdomyosarcoma (SJCRH30); triple-negative breast cancer (MDA-MB-231); two epithelial tumor cell lines (HNE-1 and HONE-1; undifferentiated lung carcinoma (Chago-K1).

**Table 3 molecules-28-04744-t003:** Clerodane diterpenes with anti-inflammatory activity.

Plant Source	Compound Name	Methods	Results	References
** *Ajuga* ** ** *pantantha* **	Ajugapantin C (**141**)	**Western Blot Analysis**	Compounds **141**, **142** and **146** downregulated iNOS and COX-2 protein levels	[57,58]
		**Docking Analysis**	Compounds **141**, **142** and **146** have strong interactions with the iNOS and COX-2 proteins	
		**Griess assay** BV-2 cells stimulated LPS	IC_50_ µM 20.2	
	Ajugapantin E (**142**)	**Griess assay** BV-2 cells stimulated LPS	IC_50_ µM 45.5.	
	Ajugapantin F (**143**)		34.0	
	Ajugapantin G (**144**)		27.0	
	Ajugapantin H (**145**)		45.0	
	Ajugapantin I (**146**)		25.8	
	Pantanpene α (**147**)	**Griess assay** BV-2 cells stimulated LPS	IC_50_ μM 65.7	
	Pantanpene B (**148**)		37.7	
	Pantanpene C (**149**)		61.7	
	Pantanpene d (**150**)		>50% inhibition at 30 μM	
	Pantanpene E (**151**)	**Griess assay** BV-2 cells stimulated LPS	IC_50_ μM 21.7	
		**Anti-inflammatory assay in zebrafish model**	The anti-inflammatory effect was confirmed	
		**Docking Analysis**	Compounds **148** and **151** have strong interactions with the iNOS and COX-2 proteins	
** *Callicarpa arborea* **	Callicarpin A (**152**)	**NLRP3 Inflammasome activation assay** J774A.1 cells were primed with LPS	IC_50_ μM 16.6	[59]
	Callicarpin B (**153**)		4.0	
	Callicarpin C (**154**)		25.4	
	(16*S*)-Tris-*O*-Acetylcallicarpin C (**155**)		5.3	
	Callicarpin E (**156**)		24.7	
	Callicarpin F (**157**)		1.5	
	Callicarpin G (**158**)	**NLRP3 Inflammasome activation assay** J774A.1 cells were primed with LPS	IC_50_ μM 1.4	
		**Pyroptosis fluorescence microscopy**	The compound **153 inhibited pyroptosis and blocked NLRP3 inflammasome activation by hampering Casp-1 cleavage and IL-1β secretion**	
** *Callicarpa cathayana* **	Cathayanalactone A (**159**)	**Griess assay** RAW264.7 macrophages stimulated LPS	IC_50_ µM 22.92	[60]
	Cathayanalactone B (**160**)		13.25	
	Cathayanalactone C (**161**)	**Griess assay** RAW264.7 macrophages stimulated LPS	IC_50_ µM 82.82	
	15-methoxypatagonic acid (**162**)		35.35	
	16-hydroxycleroda-3, 13-dien-16, 15-olide-18-oic acid (**163**)	**Griess assay** RAW264.7 macrophages stimulated LPS	IC_50_ µM 17.49	
		**ELISA assay** Quantification of TNF-α, IL-6 and IL-1β	Compounds **161**–**163** inhibited IL-1*β*, IL-6 and TNF-*α*	
** *Callicarpa hypoleucophylla* **	Callihypolin A (**164**)	**Inhibitory activities in** - superoxide anion generation and - elastase release in formyl-methionyl-leucyl-phenylalanine (fMLF)/cytochalasin (CB)-induced human neutrophils	% of inhibition 20.28 8.26	[61]
	Callihypolin B (**165**)		32.19 17.55	
	Compound **166**		31.19 12.15	
	Patagonic acid (**167**)		32.88 13.57	
	Limbatolide F (**168**)		23.65 7.33	
	Limbatolide A (**169**)		8.44 10.50	
	Compound **170**		7.93 9.30	
	Clerodermic acid (**171**)		15.23 11.80	
	Visclerodol acid (**172**)		18.80 16.30	
** *Croton crassifolius* **	Crassifolin Q (**49**)	**ELISA assay** IL-6 TNF-α	% of production 72.23 89.38	[32,62]
	Crassifolin R (**50**)		77.88 77.73	
	Crassifolin S (**51**)		73.36 79.23	
	Crassifolin T (**52**)		35.48 54.14	
	Crassifolin U (**53**)		32.78 12.53	
	Compound **173**	**Griess assay** RAW264.7 macrophages stimulated LPS	**IC_50_** μM 25.8	
	Compound **174**		**173** at 178 < 50% inhibition at 50 µM	
	C-6 epimer of crotoeuricin C (**175**)			
	Crotocaudin (**176**)			
	Teucvin (**177**)			
	Crassifolin F (**178**)			
** *Croton* ** ** *floribundus* **	Croflorin A (**179**)	**Griess assay** RAW264.7 macrophages stimulated LPS	IC_50_ μM 28.52	[63]
	Croflorin B (**180**)		40.26	
	Croflorin C (**181**)		25.47	
	Croflorin D (**182**)		35.78	
	3α-hydroxy-5,10-didehydrochiliolide (**183**)		40.58	
** *Croton laui* **	3*S*-acetoxyl-mollotucin D dilactone ester (**184**)	**Griess assay** RAW264.7 macrophages stimulated LPS	**IC_50_** µM weak activity	[64]
	6*S*-crotoeurin C (**185**)		1.2	
	Crotoeurin C (**186**)		1.6	
	Mollotucin D dilactone ester (**187**)		weak activity	
	Crassifolin F compound **178**		weak activity	
** *Croton* ** ** *poomae* **	Crotonolide K (**188**)	**Griess assay** RAW264.7 macrophages stimulated LPS	**IC_50_** µM 46.43	[65]
	Furocrotinsulolide A acetate (**189**)		31.99	
	Furocrotinsulolide A (**190**)		81.97	
	Compound **191**		86.98	
	Compound **192**		48.85	
	Crotonolide E (**193**)		74.78	
	Crotonolide F (**194**)		42.04	
	Compound **195**		32.19	
** *Dodonaea viscosa* **	Hautriwaic acid (**196**)	**Arthritis in mice** induced by caolin/carrageenan Doses mg/kg 5 10 20	% inflammation of edema after 15 days 27 20 13	[66]
		**ELISA assay** Quantification of IL-10, TNF-α, IL-6 and IL-1β	Compound **196** diminished TNF-α, IL-6 and IL-1β and increased IL-10	
** *Dysoxylum lukii.* **	neoclerod-13Z-ene-3α, 4β, 15-triol (**197**)	**Griess assay** RAW264.7 macrophages stimulated LPS	IC_50_ µM. 25.5	[67]
** *Jamesoniella* ** ** *autumnalis* **	Jamesoniellide Q (**198**)	**Griess assay** RAW264.7 macrophages stimulated LPS	IC_50_ µM 45.10	[68]
	Jamesoniellide R (**199**)		82.98	
** *Monoon membranifolium* **	2β-Methoxyhardwickiic acid (**200**)	**Griess assay** RAW264.7 macrophages stimulated LPS	**IC_50_** µM 65.4	[69]
	(-)-hardwickiic acid (**91**)		38.9	
	2β-acetoxyhardwickiic acid (**201**)		16.1	
	2β-hydroxyhardwickiic acid (**202**)		82.4	
	15-methoxypatagonic acid (**203**)		28.9	
** *Nepeta suavis.* **	Nepetolide (**204**)	**Carrageenan-induced hind paw edema** **Docking Analysis** **In silico evaluation**	Compound 204 inhibited hind paw edema Target Cox-2 EGFR and Lox-2	[70]
** *Polyalthia longifolia* **	16-oxo-cleroda-3,13(14)E-dien-15-oic acid (**205**)	**Cyclooxygenase inhibitory assay 5-LOX kit** COX-1 COX-2 5-LOX	IC_50_ µM 8.00 8.41 8.41	[40,71]
	16-hydroxy-cleroda-3,13-dien-15-oic acid (**206**)	COX-1 COX-2 5-LOX	9.75 4.07 9.78	
	16-hydroxy-cleroda-4(18),13-dien-16,15-olide (**74**)	COX-1 COX-2 5-LOX	3.77 2.71 4.06	
	3α,16α-dihydroxy-cleroda-4(18),13(14)Z-dien-15,16-olide (**75**)	COX-1 COX-2 5-LOX	3.63 4.29 5.67	
	16α-hydroxy-cleroda-3,13(14)Z-dien-15,16-olide (**76**)	COX-1 COX-2 5-LOX	3.01 3.29 4.58	
		**Docking Analysis** **In silico evaluation**	Compounds **74**–**76** have interactions with COX-1/2 and LOX enzymes	
	3β-16a-dihydroxy-cleroda-4(18),13(14)Z-dien-15,16-olide (**77**)	**ELISA assay** Quantification of cytokines such as TNF-α, TGF-β, IL-6, IL-10 and IL-1β	Compounds **74** and **77** inhibited production of proinfammatory cytokines and increased IL-10 and TGF-β	
		**Docking Analysis** **In silico evaluation**	Compound **74** docked into the active sites of MDM2, TNF-α, FAK and IL-6 Compound **77** docked into the active sites of MDM2, TNF-α, TGF-β and FAK	
** *Scutellaria* ** ** *barbata* **	Scuttenline C (**207**)	**Griess assay** RAW264.7 macrophages stimulated LPS	**IC_50_** μM 1.9	[72]
	Barbatin A (**208**)		12.6	
	Scutebarbatine F (**209**)		3.7	
** *Teucrium fructicans* **	11-hidroxyfruticolone (**210**)	**Griess assay** RAW264.7 macrophages stimulated LPS	**IC_50_** μM 39.3	[73]
** *Tinospora crispa* **	Crispinoid D (**211**)	**qPCR assay** IL-1β, IL-6, TNF-α, iNOs, CCL12 and COX-2	Compounds **211**–**213** diminish the production of pro-inflammatory mediators	[74,75]
		**Luciferase assay:** Inhibition of NF-κB	**IC_50_** μM 5.94	
	Tinosporol C (**212**)	Inhibition of NF-κB	6.32	
	marrubiagenin-methylester (**213**)	Inhibition of NF-κB	25.20	
	Tinopanoid A (**214**)	**Griess assay** BV-2 cells stimulated LPS	**IC_50_** μM >60	
	Tinopanoid B (**215**)		>60	
	Tinopanoid C (**216**)		24.1	
	Tinopanoid D (**217**)		41.1	
	Tinopanoid E (**218**)		7.5	
	Tinopanoid F (**219**)		50.8	
	Tinopanoid G (**220**)		10.6	
	Tinopanoid H (**221**)		39.4	
	Tinopanoid I (**222**)		59.1	
	Tinopanoid J (**223**)		45.9	
	Tinospin C (**224**)		>60	
	borapetol B (**225**)		>60	
	Tinotufolin D (**226**)		14.5	
** *Tinospora sagittata* **	Fibaruretin H (**227**)	**Griess assay** RAW264.7 macrophages stimulated LPS	% inhibition at 24 µM 27.0%	[76]
	Fibaruretin I (**228**)		33.1%	

Compound **166** (4*aR*,5*S*,6*R*,8a*R*)-5-[2-(2,5-dihydro-5-methoxy-2-oxofuran-3-yl)ethyl]-3,4,4a,5,6,7,8,8a-octahydro-5,6,8a-trimethylnaphthalene-1-carboxylic acid); Compound **170** (methyl (4*aR*,5*S*,6*R*,8*S*,8*aR*)-3,4,4a,5,6,7,8,8a-octahydro-8-hydroxy-5,6,8a-trimethyl-5-[2-(2-oxo-2,5-dihydrofuran-3-yl)ethyl]naphthalene-1-carboxylate); Compound **173** (3*S*,4*S*,6*S*,8*R*,9*R*,12*S*)-3-acetoxy-18-methoxycarbonyl-6,19:15,16-diepoxy-halim-5(10),13(16),14-triene-20,12-olide; Compound **174** (3*S*,4*S*,6*S*,8*R*,9*R*,12*S*)-3,19-diacetoxy-18-methoxycar-bonyl-15,16-epoxy-6-hydroxyhalim-5(10),13(16),14-triene-20,12-olide; Compound **191** (3,4,15,16-diepoxycleroda-13(16),14-diene-12,17-olide); Compound **192** (15,16-epoxy-3β-hydroxy-5(10),13(16),14-dien-12,17-olide; Compound **195** (3β,4β:15,16-diepoxy-13(16),14-clerodadiene; Compound **226** (2aβ,3α,5aβ,6β,7α,8aα)-6-[2-(3-furanyl)ethyl]-2a,3,4,5,5a,6,7,8,8a,8b-decahydro-2a,3-dihydroxy-6,7,8b-trimethyl-2H-naphtho[1,8-bc]furan-2-one). Cells are immortalized by v-raf/v-myc carrying J2 retrovirus (BV-2); inducible nitric oxide synthase (iNOS); cyclooxygenase 2 (COX-2); key sensor molecule in the inflammasome activity (NLRP3); protein found on the surface of some cells that binds epidermal growth factor (EGFR); 5-lipoxigenasa (5-LOX); tumor necrosis factor-α (TNF-α); interleukin-6 (IL-6); interleukin 1β (IL-1β); proinflammatory–chemokine (C-C motif) ligand 12 (CCL12).

## Data Availability

Not applicable.

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
