# Peer review of "Anti-Inflammatory and Cytotoxic Activities of Clerodane-Type Diterpenes"

_molecules, 2023, doi:10.3390/molecules28124744_

Round 1
Reviewer 1 Report
The authors undertake 8 years of study on clerodane and neo-clerodane diterpenes that exhibit cytotoxic and anti-inflammatory activities. The research was based on the databases PubMed, Google Scholar, and Science Direct. Their results showed that clerodane diterpenes have activity against different cell cancer lines. Yet, some diterpenes presented in this review have shown therapeutic targets.
The authors used extended databases for collecting their results followed by an in-depth discussion. Their results were collected in a detailed table that reports the methodology used and its effect.
It is an interesting study that will reinforce scientists to proceed with more research in the field. There is a rich bibliography on these subjects.
My suggestion is to ACCEPT and publish the article
Author Response
Reviewer 1.
Thank you for your comments.
Reviewer 2 Report
Review of Molecules 2414786 Anti-inflammatory and cytotoxic activities of clerodanes-type diterpenes
The complexity of the Clerodanes is daunting but in keeping with the biodiversity of phytochemicals in general. This submission is an extensive listing of Clerodane diterpene compounds and is well presented. I would have liked comments on their areas of therapeutic application in a more expanded format and information on their relative toxicities also provided. This is the type of information that the readers would find particularly useful. What are the solubility properties of Clerodanes, are water soluble derivatives available.
For example the related bicyclic tetrahydropyran ring containing labdane diterpene forskolin has many interesting properties as a plant stimulant for the production of cyclic AMP by mammalian cells through induction of adenylate cyclase which has been applied in the treatment of bronchoconstriction and heart failure and as a folkloric medicine as a heart tonic. Although relatively poorly researched Forskolin has nevertheless been used as a dietary supplement for the treatment of obesity due to its ability to act as an appetite suppressant and ability to stimulate lipase and adenylate cyclase enzymatic activities. Clearly further research is required on forskolin but the information presently at hand at least is at least a pointer to areas where it may find therapeutic application and provides an impetus for further investigation. Based on the information provided in the current submission practical information on the Clerodane’s can be improved for the submission to serve a similar role. If relatively little knowledge is currently available on the Clerodanes, the authors should indicate this but may also speculate on the potential of certain compounds based on what is known as it relates to specific areas of therapeutic application of potential emerging areas in biomedicine. Have derivatives been prepared of the Chlorodanes with improved efficacy and solubility?
Minor point
Ensure that the size of numbering is consistent through each figure.
Author Response
Reviewer 2.
The complexity of the Clerodanes is daunting but in keeping with the biodiversity of phytochemicals in general. This submission is an extensive listing of Clerodane diterpene compounds and is well presented. I would have liked comments on their areas of therapeutic application in a more expanded format and information on their relative toxicities also provided. This is the type of information that the readers would find particularly useful.
Answer: Only clerodanes 75 and 77 have acute toxicity, but, it is not mentioned LD50.
For example the related bicyclic tetrahydropyran ring containing labdane diterpene forskolin has many interesting properties as a plant stimulant for the production of cyclic AMP by mammalian cells through induction of adenylate cyclase which has been applied in the treatment of bronchoconstriction and heart failure and as a folkloric medicine as a heart tonic. Although relatively poorly researched Forskolin has nevertheless been used as a dietary supplement for the treatment of obesity due to its ability to act as an appetite suppressant and ability to stimulate lipase and adenylate cyclase enzymatic activities. Clearly further research is required on forskolin but the information presently at hand at least is at least a pointer to areas where it may find therapeutic application and provides an impetus for further investigation. Based on the information provided in the current submission practical information on the Clerodane’s can be improved for the submission to serve a similar role. If relatively little knowledge is currently available on the Clerodanes, the authors should indicate this but may also speculate on the potential of certain compounds based on what is known as it relates to specific areas of therapeutic application of potential emerging areas in biomedicine
Answer: The compounds that presented greater cytotoxic and/or anti-inflammatory activity were discussed, and we speculate about their potential therapeutic application
What are the solubility properties of Clerodanes, are water soluble derivatives available.
Answer: This review only included clerodanes isolated from plants but not derivatives, so we do not have this information. However, we think that these compounds are not water soluble.
Reviewer 3 Report
The paper entitled "Anti-inflammatory and cytotoxic activities of clerodanes-type diterpenes" constitute an interesting review with knowledge on this topic.
Author Response
Reviewer 3
Thank you for your comments
Reviewer 4 Report
In this manuscript, Julia Pérez-Ramos listed the clerodanes and neo-clerodanes with cytotoxic or anti-inflammatory activity from 2015 to February 2023. This work is somewhat interesting, but there are some major concerns as followings:
1. There are many reviews about the clerodane diterpenes, including the reviews cited in this manuscript ([1], [2] and [3]). And the topics of some reviews are closely related to that of this manuscript. What’s novelty of your review?
2. The two schemes should be merged into one, as both of them shared the same starting compound and some intermediates.
3. Some structures were wrong including the missing groups and the unclear configurations. For instances, the OPP group was missing for the second intermediate in the first line of Figure 1, the methyl group was missing for the forth skeleton in the Figure 2. Please check the structures throughout the whole manuscript.
4. What are the names of the families of these plants? Which plants belonging to the family Lamiaceae? Where do these plants grow? And which parts of plants were these diterpenes isolated from? Only two tables were insufficient. Authors should provide textual descriptions as part of the main text.
5. Which diterpenes belonging to clerodanes or neo-clerodanes? In addition, authors listed four types of clerodane skeletons, but they didn’t point out which diterpenes belonging to which skeletons.
6. The names of some compounds need to be revised. For instances, ‘Corymbulosins X (9)’ should be revised as ‘Corymbulosin X (9)’, the dot in the name ‘2R,5S,6S,8R,9R,10S,18S,19S)-2,19-diacetoyloxy-6,18-di-methoxy-18,19-epoxycleroda-3,13(16),14-triene. (24)’ should be deleted and the left parenthesis was missing.
7. P4: Please delete ‘(SI)’ in the table on P4, according to Ref. [22].
8. For a review, it is important to give summaries on the structure-activity relationships in the Discussion section, apart from the repeat of the literature.
9. Please update the information for Ref. [22] and [27].
There are quite a lot of typo or grammar errors to be corrected, such as ‘according to their structure and carbon chain’ (P1L30), ‘has attracted interest’ (P3L83), ‘was found summarizes’ (P1L30) in the main text, even in the captions of figures (‘Figure 2. Types of clerodanes skeleton’).
Author Response
Reviewer 4
In this manuscript, Julia Pérez-Ramos listed the clerodanes and neo-clerodanes with cytotoxic or anti-inflammatory activity from 2015 to February 2023. This work is somewhat interesting, but there are some major concerns as followings:
There are many reviews about the clerodane diterpenes, including the reviews cited in this manuscript ([1], [2] and [3]). And the topics of some reviews are closely related to that of this manuscript. What’s novelty of your review?
Answer: The previous reviewers were published before 2016, and in the present review are included clerodanes isolated after this year.
- The two schemes should be merged into one, as both of them shared the same starting compound and some intermediates.
Answer: Figures 1 and 2 were merged into one.
- Some structures were wrong including the missing groups and the unclear configurations. For instances, the OPP group was missing for the second intermediate in the first line of Figure 1, the methyl group was missing for the forth skeleton in the Figure 2. Please check the structures throughout the whole manuscript.
Answer: All structures were reviewed and corrected.
- What are the names of the families of these plants? Which plants belonging to the family Lamiaceae? Where do these plants grow? And which parts of plants were these diterpenes isolated from? Only two tables were insufficient. Authors should provide textual descriptions as part of the main text.
Answer: We included table 1 with family, parts of the plants from the diterpenes were isolated and the collected place.
- Which diterpenes belonging to clerodanes or neo-clerodanes? In addition, authors listed four types of clerodane skeletons, but they didn’t point out which diterpenes belonging to which skeletons.
Answer: We included which compounds clerodanes and neo-clerodanes in the lines 80 and 81.
- The names of some compounds need to be revised. For instances, ‘Corymbulosins X (9)’ should be revised as ‘Corymbulosin X (9)’, the dot in the name ‘2R,5S,6S,8R,9R,10S,18S,19S)-2,19-diacetoyloxy-6,18-di-methoxy-18,19-epoxycleroda-3,13(16),14-triene. (24)’ should be deleted and the left parenthesis was missing.
Answer: All the names were reviewed and corrected
- P4: Please delete ‘(SI)’ in the table on P4, according to Ref. [22].
Answer: We defined SI (selective index) in the end of table 2.
- For a review, it is important to give summaries on the structure-activity relationships in the Discussion section, apart from the repeat of the literature.
- Please update the information for Ref. [22] and [27].
Answer: The corrections indicated in 8 and 9 comments were made.
Round 2
Reviewer 4 Report
The manuscript was improved by the authors. It is recommended to be accepted after minor revisions.
1. In order to give readers a better understanding of the biogenetical pathways of clerodanes, the description should be modified according to Ref. [3].
2. The pages for Ref. [22] and [27] were still missing in the revised manuscript.
Author Response
Ms. Ewa Myszka, M.Sc.
Assistant Editor
Thank you for giving us the opportunity to submit a revised draft of the manuscript “Anti-inflammatory and cytotoxic activities of clerodanes-type diterpenes” for publication in the Molecules.
We appreciate the time and effort that you and the reviewers dedicated to providing feedback on our manuscript and are grateful for the insightful comments on and valuable improvements to our paper.
We have included the suggestions of the reviewer 4. The changes were marked in red color, such that any modifications can be easily viewed by the editor and reviewer.
Reviewer 4f
In order to give readers a better understanding of the biogenetical pathways of clerodanes, the description, should be modified according to Ref (3).
Answer:
Figure 1 was improved according to Ref (3)
- The pages for Ref (22) and (27) were still missing in the revised manuscript
Answer: Te pages of Ref (22) and (27) were included.
Regard
Julia Pérez Ramos